∂ | **Open Peer Review** | Antimicrobial Chemotherapy | Methods and Protocols

# Assessment of the revision of the 2025 CLSI breakpoints for the interpretation of minocycline susceptibility for *Acinetobacter baumannii* complex

Xiaochen Yu,[1,2] Yanhua Liu,[1,3] Jinye Du,[1] Fupin Hu,[1] Dandan Yin[1]

**ABSTRACT** To evaluate the revision to the 2025 Clinical and Laboratory Standard Institute (CLSI) breakpoints revision on the interpretation of minocycline for *Acinetobacter baumannii* complex, by comparing the minimum inhibitory concentrations obtained by broth microdilution (BMD) with inhibition zone diameters measured by disk diffusion testing. A total of 276 non-duplicate clinical isolates of *A. baumannii* complex were collected in China between 2022 and 2024, including 150 carbapenem-resistant *A. baumannii* (CRAB) isolates and 126 carbapenem-susceptible *A. baumannii* isolates. Antimicrobial susceptibility testing was performed using BMD and disk diffusion, with results interpreted per the 2024 and 2025 breakpoints, respectively. Categorical agreement and error rates were calculated to assess performance. Compared to the 2024 CLSI breakpoints, the updated 2025 CLSI breakpoints reduced the susceptibility rate of *A. baumannii* complex to minocycline (from 73.9% to 46.4%) and increased the resistance rate (from 4.4% to 46.0%). For CRAB, the susceptibility rate dropped from 53.3% to 6.0%, while the resistance rate rose from 7.4% to 82.7%. Categorical agreement improved from 64.1% to 90.9%. The intermediate results proportion determined by disk diffusion decreased significantly (from 25.4% to 5.8%). Notably, 62.5% of isolates categorized as intermediate by disk diffusion were reclassified as either susceptible or resistant when tested by BMD. The revised 2025 CLSI breakpoints are clinically applicable. They reduce the susceptibility rate and increase the resistance rate of *A. baumannii* complex to minocycline. There is also improved categorical agreement and reduced error rates between disk diffusion and BMD. For isolates categorized as intermediate by disk diffusion, confirmation with BMD is recommended.

**IMPORTANCE** The Clinical and Laboratory Standards Institute (CLSI) has made a comprehensive revision of minocycline breakpoints for *Acinetobacter* species in 2025. This update may have a significant impact on antimicrobial susceptibility testing interpretation and treatment strategies for clinical laboratories. However, there is currently a lack of comparative studies about it. Therefore, this study aims to evaluate the impact of the 2025 CLSI breakpoints revision on the interpretation of minocycline for *Acinetobacter baumannii* complex and to provide experimental evidence for the clinical applicability of new breakpoints.

**KEYWORDS** minocycline, breakpoint, susceptibility testing, *Acinetobacter baumannii*

**Editor** Andrea M. Prinzi, bioMerieux Inc, Salt Lake City, Utah, USA

**Peer Reviewer** Ayesha Khan, UCI Health, Orange, California, USA

Address correspondence to Fupin Hu, hufupin@fudan.edu.cn, or Dandan Yin, yyindandan@163.com.

Xiaochen Yu and Yanhua Liu contributed equally to this article. The order of co-first-author names was determined by the first letter.

The authors declare no conflict of interest.

See the funding table on p. 8.

*A*cinetobacter baumannii complex is a common pathogen in clinical settings, causing ventilator-associated pneumonia, catheter-related infections, and skin and urinary tract infections (1). According to data from the China Antimicrobial Surveillance Network (CHINET), *A. baumannii* complex was the fifth most common bacterial pathogens isolated from clinical specimens, with a carbapenem resistance rate of up to 60% (2).

The same data on carbapenem-resistant *A. baumannii* (CRAB) from the intensive care units in southwestern China in 2018–2022 remained above 70% (3). Due to its multidrug resistance and the limited number of effective treatment options, CRAB has been designated as a high-priority pathogen by major global public health authorities (4, 5).

Minocycline is one of the antimicrobial agents recommended by the Infectious Diseases Society of America (IDSA) for treating CRAB (6) and has attracted considerable attention for the treatment of *A. baumannii* complex infections due to its favorable tissue penetration and intracellular activity. As the IDSA guidance outlined, for CRAB infections treatment, the preferred regimen is sulbactam-durlobactam combined with imipenem-cilastatin or meropenem. If sulbactam-durlobactam is unavailable, an alternative option is high-dose ampicillin-sulbactam combined with at least one additional active agent, such as polymyxin B, minocycline (prior to tigecycline), or cefiderocol. However, studies have shown that the susceptibility results of minocycline are significantly influenced by the testing method used, with a risk of discordance observed between the disk diffusion (DD), E-test, Vitek2, and broth microdilution (BMD) methods. For example, a study for 107 CRAB clinical isolates showed that very major errors were rare, but major and minor errors overcalling strains as intermediate or resistant occurred frequently with susceptibility testing methods that were feasible in clinical laboratories (7). Another study also showed that both Vitek2 and Etest would produce higher minocycline MICs compared with the reference method (8). This poses a challenge for consistent and accurate categorical interpretation in clinical laboratories. Tsakris et al. used Monte Carlo simulations to suggest that the previous "MIC ≤ 4 mg/L" susceptibility breakpoint may overestimate the clinical effectiveness of minocycline. They indicated that the conventional dosage could only achieve pharmacokinetic/pharmacodynamic (PK/PD) targets when the MIC was ≤1 mg/L, highlighting the need to revise the susceptibility breakpoints based on PK/PD parameters and clinical outcome data (9). Similar recommendations have been made in Taiwan studies, which suggest that the minocycline susceptibility breakpoint should be lowered to 1 mg/L to more accurately reflect the *in vivo* therapeutic potential (10).

The Clinical and Laboratory Standards Institute (CLSI) has comprehensively revised the minocycline breakpoints for *Acinetobacter* spp., as shown in Table 1 (11). This update could significantly impact the interpretation of antimicrobial susceptibility testing and treatment strategies in clinical laboratories. However, comparative studies on this topic are currently lacking. This study, therefore, aims to evaluate the impact of the 2025 CLSI breakpoints revision on the interpretation of minocycline for *A. baumannii* complex and to provide experimental evidence of the clinical applicability of the new breakpoints.

## MATERIALS AND METHODS

### Clinical isolates

A total of 276 non-duplicate *A. baumannii* complex clinical isolates, including 150 CRAB and 126 carbapenem-susceptible *A. baumannii* (CSAB), were collected from 47 hospitals in 23 provinces between January 2022 and October 2024 through the CHINET (https://www.chinets.com/). Isolate identification was confirmed using the matrix-assisted laser desorption ionization-time-of-flight mass spectrometry (bioMérieux, France). In

**TABLE 1** Comparison of zone diameter and MIC breakpoints for minocycline in the 34th (2024) and 35th (2025) edition of the CLSI M100 guideline

| CLSI 100 of different editions | Breakpoints | | | | | |
|---|---|---|---|---|---|---|
| | MIC (mg/L) | | | Zone diameter (mm) | | |
| | S | I | R | S | I | R |
| 35th (2025) | ≤1 | 2 | ≥4 | ≥22 | 18–21 | ≤17 |
| 34th (2024) | ≤4 | 8 | ≥16 | ≥16 | 13–15 | ≤12 |

this study, CRAB was defined as *A. baumannii* complex isolates that were resistant to imipenem by BMD, and CSAB was imipenem-susceptible isolates.

## Antimicrobial susceptibility testing

Antimicrobial susceptibility testing was performed using BMD (0.06–128 mg/L) with cation-adjusted Mueller-Hinton broth (Lot 1079890, BD) and DD (30 µg, Lot 3714739, OXOID) (using the same bacterial suspension) according to the CLSI M100 guidelines. Quality control strains *Escherichia coli* ATCC 25922 were tested in each testing experiment with clinical strains, for both the BMD and DD methods. Additionally, a rigorous quality control and assurance protocol was implemented in this study, including the adoption of standardized operating procedures and regular equipment calibration. In addition, the reading of experimental results in each assay was performed and cross-verified by two independent operators. For isolates with discrepant results in the initial susceptibility testing, all testing would be repeated, and the repeat results would be used in the final analysis, regardless of whether they differed from initial results.

## Data analysis

The evaluation indices included categorical agreement (CA), very major error (VME), major error (ME), and minor error (mE) rates. CA was defined as the percentage of isolates that were categorized as susceptible, intermediate, or resistant according to the respective breakpoints of the DD and BMD methods. VME referred to the percentage of isolates that were classified as resistant by BMD but susceptible by DD (the denominator was the number of resistant isolates). ME referred to isolates deemed susceptible by BMD but resistant by DD (the denominator was the number of susceptible isolates), and mE referred to isolates classified as intermediate by BMD but as either susceptible or resistant by DD. According to the CLSI M52 guidelines (12), the acceptable thresholds for CA, VME, ME, and mE are ≥90%, ≤1.5%, ≤3%, and ≤10%, respectively.

## RESULTS

### The antimicrobial activity of minocycline against *A. baumannii* complex

The $MIC_{50}$ and $MIC_{90}$ of minocycline against 276 *A. baumannii* complex isolates were 2 mg/L and 8 mg/L, respectively, with an MIC range of ≤0.06–16 mg/L. For CRAB ($n$ = 150) and CSAB ($n$ = 126), the $MIC_{50}$/$MIC_{90}$ were 4/8 mg/L and 0.125/2 mg/L, respectively (Table 2). Compared to the 2024 CLSI breakpoints (13), the susceptibility rate to minocycline decreased from 73.9% to 46.4%, while the resistance rate increased from 4.4% to 46.0%, and the intermediate rate decreased from 21.7% to 7.6%, when the updated 2025 breakpoints were used. For CRAB, the susceptibility rate fell sharply from 53.3% to 6.0%, the resistance rate rose from 7.4% to 82.7%, and the intermediate rate fell from 39.3% to 11.3%. By contrast, CSAB isolates showed a modest change: the susceptibility rate decreased from 98.4% to 94.4%, the resistance rate increased from 0.8% to 2.4%, and the intermediate rate rose from 0.8% to 3.2% (Table 2).

Regarding the DD results, the overall susceptibility rate to minocycline decreased from 54.3% to 45.3% following the updated interpretation. Meanwhile, the resistance rate increased from 20.3% to 48.9%, and the intermediate rate fell from 25.4% to 5.8%. For CRAB, the susceptibility rate fell from 18.0% to 3.3%, the resistance rate rose from 36.7% to 88.0%, and the intermediate rate fell markedly from 45.3% to 8.7%. For CSAB, there was a slight decrease in the susceptibility rate from 97.6% to 95.2%, an increase in the resistance rate from 0.8% to 2.4%, and an increase in the intermediate rate from 1.6% to 2.4% (Table 2).

### Comparison between DD and BMD

We also examined the consistency between the DD method and the reference BMD for minocycline susceptibility based on different CLSI breakpoints (Fig. 1A through C). By the

**TABLE 2** Distribution of MICs and shifts in susceptibility in the 34th (2024) and 35th (2025) edition of the CLSI M100 guideline

| Parameter | All isolates | CSAB | CRAB |
|---|---|---|---|
| No. of isolates | 276 | 126 | 150 |
| MIC (mg/L) | | | |
| Range | ≤0.06–16 | ≤0.06–16 | 0.125–16 |
| $MIC_{50}$ | 2 | 0.125 | 4 |
| $MIC_{90}$ | 8 | 2 | 8 |
| % of isolates (MIC) | | | |
| 35th (2025) | | | |
| S | 46.4 (128) | 94.4 (119) | 6.0 (9) |
| I | 7.6 (21) | 3.2 (4) | 11.3 (17) |
| R | 46.0 (127) | 2.4 (3) | 82.7 (124) |
| 34th (2024) | | | |
| S | 73.9 (204) | 98.4 (124) | 53.3 (80) |
| I | 21.7 (60) | 0.8 (1) | 39.3 (59) |
| R | 4.4 (12) | 0.8 (1) | 7.4 (11) |
| % of isolates (inhibition zone diam) | | | |
| 35th (2025) | | | |
| S | 45.3 (125) | 95.2 (120) | 3.3 (5) |
| I | 5.8 (16) | 2.4 (3) | 8.7 (13) |
| R | 48.9 (135) | 2.4 (3) | 88.0 (132) |
| 34th (2024) | | | |
| S | 54.3 (150) | 97.6 (123) | 18.0 (27) |
| I | 25.4 (70) | 0 (0) | 45.3 (68) |
| R | 20.3 (56) | 2.4 (3) | 36.7 (55) |

2025 breakpoints, the CA, VME, ME, and mE rates were 90.9% (251/276), 0% (0/127), 0% (0/128), and 5.4% (15/276), respectively. These rates all meet the acceptable thresholds recommended in the CLSI M52 document (12). However, using the 2024 breakpoints on the same data set yielded different results: CA was 64.1% (177/276), VME was 0% (0/12), ME was 2.5% (5/204), and mE was 15.2% (42/276). Here, the CA and mE rates failed to meet the recommended criteria (Table 3). Subgroup analysis for CRAB revealed substantial differences in CA (34.7% vs 85.3%), mE (28.0% vs 8.7%), ME (6.3% vs 0%), and VME (0% vs 0%) for DD testing, according to the 2024 and 2025 breakpoints. For CSAB isolates, the results were largely consistent, with CA (97.6% vs 98.4%), mE (1.6% vs 0.8%), ME (0% vs 0%), and VME (0% vs 0%), showing minimal differences between the two breakpoints (Table 3).

Further analysis revealed a clear correlation between minocycline MIC values and inhibition zone diameters (Fig. 2). Among the susceptible isolates (MIC ≤ 1 mg/L), 96.1% (123/128) exhibited inhibition zones within the susceptible range (22–34 mm). Similarly, 96.1% (122/127) of the resistant isolates (MIC ≥ 4 mg/L) showed resistant zone diameters (7–17 mm). In contrast, isolates with intermediate MICs (MIC = 2 mg/L) displayed a more scattered distribution in the DD results. 28.6% (6/21) fell within the intermediate zone (18–21 mm), while 61.9% (13/21) had zone diameters of ≤17 mm and would be misclassified as resistant using the DD method. 9.5% (2/21) showed zones ≥22 mm and would be misclassified as susceptible. In this study, a total of 15 isolates exhibited mE, all with an MIC of 2 mg/L, which was interpreted as intermediate by BMD. Thirteen of these isolates had inhibition zone diameters of <17 mm and were categorized as resistant (Fig. 3A and B), while the remaining two isolates showed diameters of 30 mm and were categorized as susceptible by DD.

## Special result interpretation

Minocycline susceptibility testing generally produces clear results. In BMD testing, the MIC endpoints are well-defined. In DD testing, the inhibition zones often show sharply

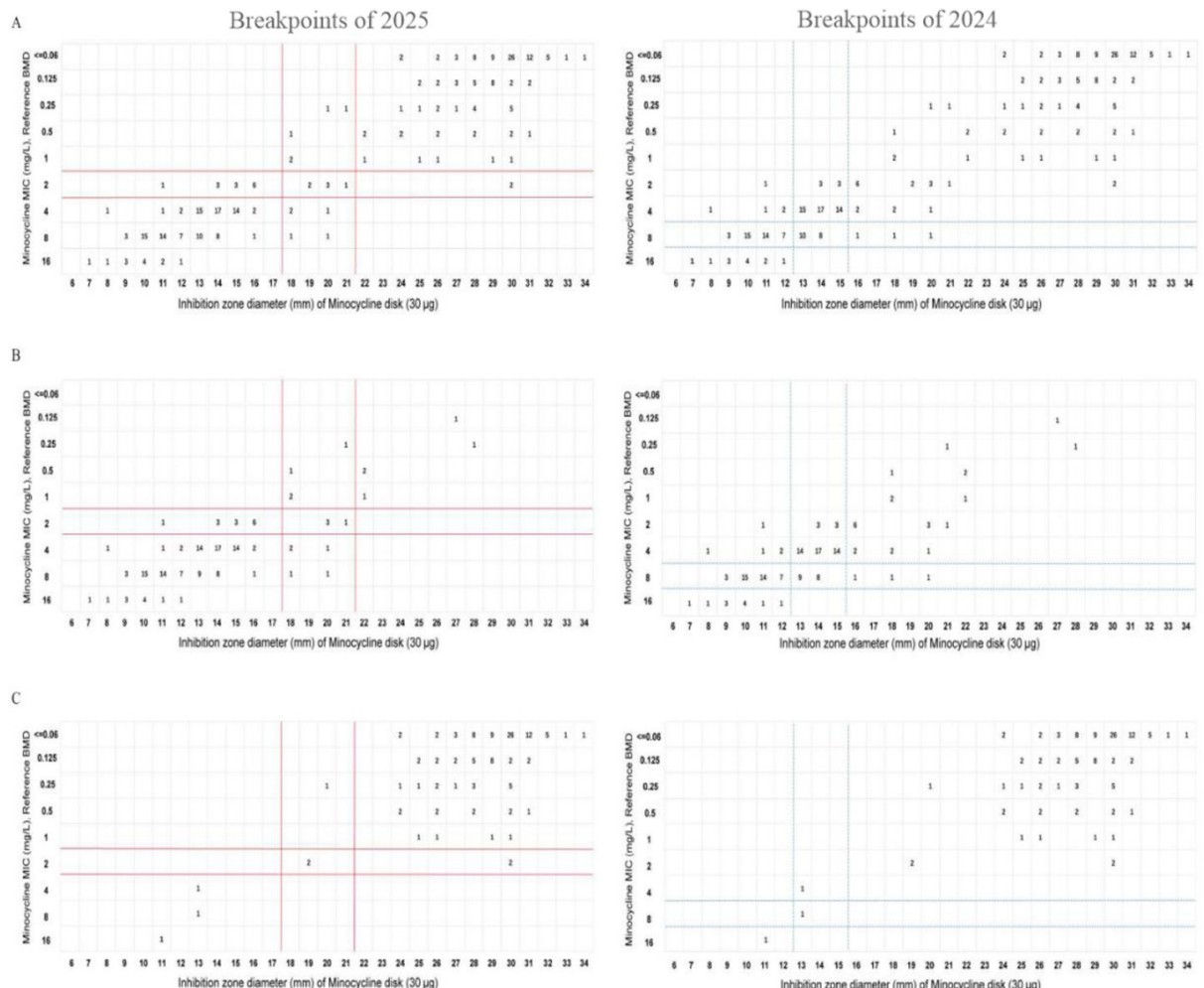

**FIG 1** Scattergrams showing the distribution of MICs and inhibition zone diameters for minocycline against 276 *A. baumannii complex* isolates using different breakpoints. The red solid line shows the minocycline breakpoints in the 35th (2025) edition of the CLSI M100 guideline, and the blue dotted line shows the breakpoints in the 34th (2024) edition. Panels A–C show the scattergrams for all isolates (*n* = 276), carbapenem-resistant isolates (*n* = 150), and carbapenem-susceptible isolates (*n* = 126), respectively.

defined ("cliff-like") edges, which facilitates the accurate measurement of the zone diameters (Fig. 4A). However, a small number of isolates exhibited irregular inhibition zone shapes (Fig. 4B), which can result in inaccurate diameter measurements.

## DISCUSSION

The global burden of CRAB infection remains severe, with high mortality, substantial healthcare costs, and rising prevalence in healthcare settings (14, 15). Despite CRAB's multi-drug resistance, minocycline retains clinical value: two Chinese studies support its role as an alternative to combination therapy for CRAB treatment (1, 16), and over half of polymyxin-resistant CRAB isolates remain susceptible to it (17). And it exhibits lower resistance rates (54.2%) in hypervirulent CRAB (hv-CRAB), which would lead to more severe clinical outcomes, compared to non-hv-CRAB (100%, *P* = 0.03) (16). Tetracycline-based regimens (particularly minocycline + polymyxins) have also been associated with reduced 30-day mortality in CRAB bloodstream infections (18). Another systematic review reported 72.6% clinical success and 60.2% microbiological eradication rates with minocycline (mono/combination therapy, with 100 mg or 200 mg) (19). These findings

**TABLE 3** CA and error rates for DD compared with BMD

| Organisms (no. of isolates) | Agreement or error rates (%) | | | | | | | |
|---|---|---|---|---|---|---|---|---|
| | Interpreted by 2025 CLSI | | | | Interpreted by 2024 CLSI | | | |
| | CA | mE | ME | VME | CA | mE | ME | VME |
| All isolates (276) | 90.9 (251) | 5.4 (15) | 0 | 0 | 64.1 (177) | 15.2 (42) | 2.5 (5) | 0 |
| CSAB (126) | 97.6 (123) | 1.6 (2) | 0 | 0 | 98.4 (124) | 0.8 (1) | 0 | 0 |
| CRAB (150) | 85.3 (128) | 8.7 (13) | 0 | 0 | 34.7 (52) | 28.0 (42) | 6.3 (5) | 0 |

underscore minocycline's relevance as a therapeutic option, particularly in Asia where carbapenem resistance in *A. baumannii* complex reaches 76.2% (20).

Accurate antimicrobial susceptibility testing is key to ensuring the rational use of minocycline, as prior data showed minocycline MICs a gradual distribution of the 2–8 mg/L range, closely overlapping with historical CLSI breakpoints. Consequently, even minor variations in testing could lead to shifts in the interpretation of susceptibility categories (S/I/R) (7). Our study demonstrates that the 2025 CLSI breakpoint revision addresses this challenge: it reduced the proportion of CRAB isolates in the critical MIC zone (from 46.0% to 30.1%) via BMD and significantly improved agreement between DD and BMD (CA increased from 34.7% to 85.3%; ME decreased from 6.3% to 0%). Notably, false-resistant results (ME) were eliminated in CRAB, a key improvement given minocycline's role as a salvage therapy.

The 2025 CLSI M100 also highlights the need for confirmatory BMD testing for DD inhibition zones of 18–21 mm to avoid misclassification. In our study, 5.8% of isolates fell within this range (81.3% CRAB), with only 37.5% truly intermediate—62.5% were misclassified as susceptible/resistant by DD. Additionally, about 3% of isolates exhibited mild DD inhibition zone distortion, with one strain's zone diameter precisely at the breakpoint, further justifying confirmatory BMD for borderline zones. These observations align with Wang et al. (7)'s finding that minocycline diffusion is sensitive to agar conditions, emphasizing the importance of adhering to CLSI's cautionary recommendations.

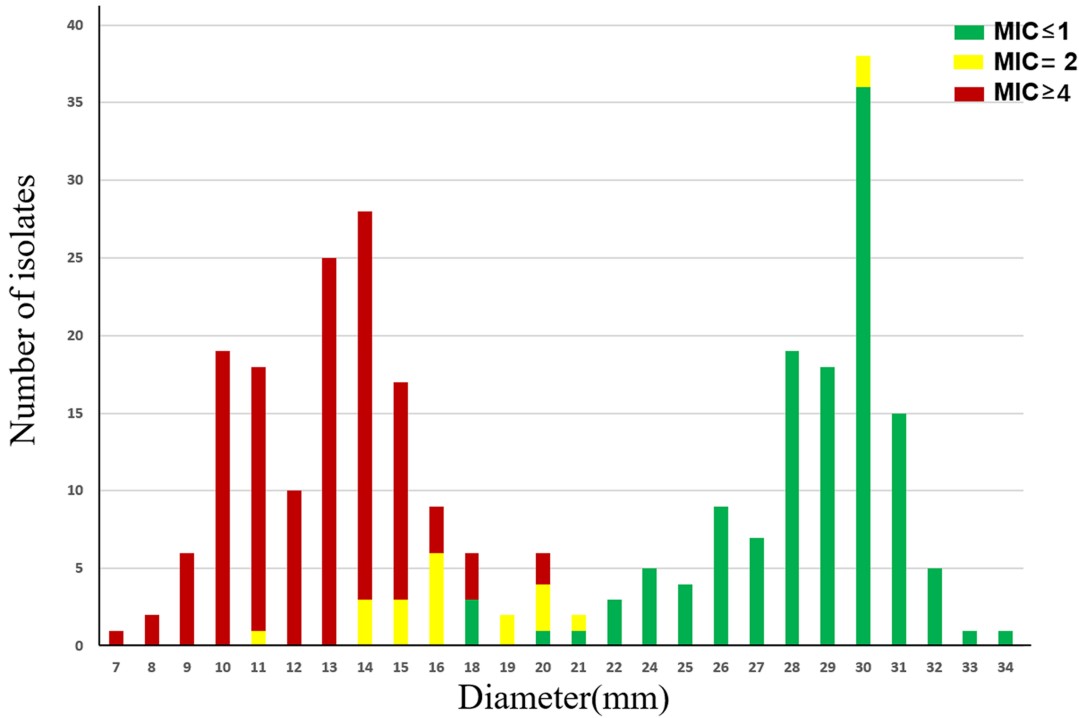

**FIG 2** Minocycline's MICs and DD inhibition zone distribution against 276 *A. baumannii* complex.

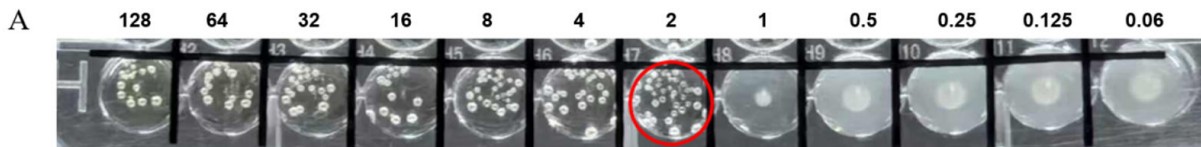

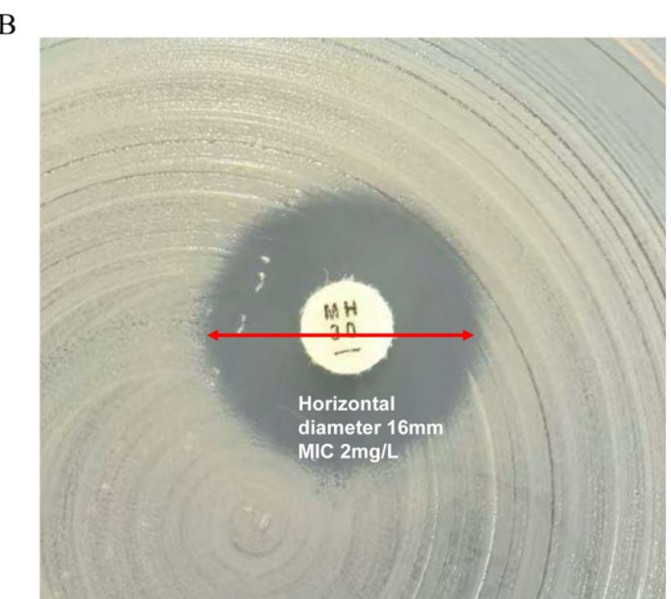

**FIG 3** A representative example of mE due to discrepancies in interpretation between methods. (A) The BMD-determined MIC was 2 mg/L (red circle), interpreted as intermediate (I). (B) The inhibition zone diameter by the DD was 16 mm, interpreted as resistant (R).

CLSI breakpoint revision follows rigorous criteria (CLSI M23) to ensure alignment with clinical efficacy and patient safety (21), as demonstrated by prior updates for piperacillin-tazobactam, colistin, and fluoroquinolones. The 2025 minocycline revision narrows interpretive ambiguity, improves the antimicrobial susceptibility testing (AST) method agreement, and supports rational use, critical for individual patient management and curbing CRAB's clonal spread.

Compared with previous single-center studies, in our study, 276 *A. baumannii* complex isolates included were collected from multiple regions, providing a certain degree of multicenter representativeness due to their diverse geographic distributions

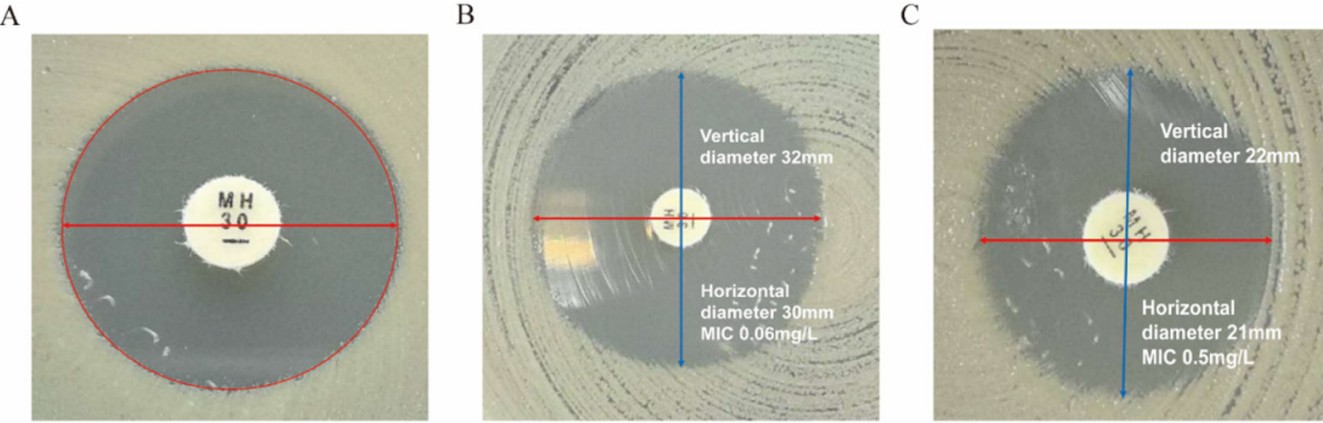

**FIG 4** Examples of atypical inhibition zone morphology affecting interpretation. (A) An inhibition zone with a sharp "cliff-like" edge; (B) an irregularly shaped inhibition zone; (C) irregular zone morphology that complicates result interpretation.

and resistance backgrounds, but the susceptibility testing was still performed at a single laboratory, so the findings of this study may therefore have certain limitations. Additionally, several other limitations should be also acknowledged. Only one brand of antibiotic discs, one type of Mueller-Hinton agar, and one type of cation-adjusted Mueller-Hinton broth were used, which may affect the generalizability of the results. Future studies should include products from multiple manufacturers to validate these findings further.

In conclusion, the 2025 CLSI minocycline breakpoints improve the accuracy and clinical utility of AST for *A. baumannii* complex, particularly CRAB. Laboratories should adopt these updated breakpoints, prioritize confirmatory BMD for borderline DD zones, and maintain strict quality control to ensure reliable susceptibility reporting and optimize minocycline use in CRAB infections.

## ACKNOWLEDGMENTS

This work was supported by the CHINET (Independent Medical Grants from Pfizer, 2020QD049), and the Shanghai Antimicrobial Surveillance Network (3030231003). The funder had no role in study design, data collection, analysis, publishing decisions, or manuscript preparation.

## AUTHOR AFFILIATIONS

[1]Institute of Antibiotics, Huashan Hospital, Fudan University, Shanghai, Shanghai, China
[2]Department of Laboratory Diagnostics, The First Affiliated Hospital of Harbin Medical University, Harbin, China
[3]Department of Clinical Laboratory, The Second Affiliated Hospital of Nanchang University, Nanchang, China

## AUTHOR ORCIDs

Fupin Hu  http://orcid.org/0000-0002-4493-0619
Dandan Yin  http://orcid.org/0000-0003-0417-158X

## FUNDING

| Funder | Grant(s) | Author(s) |
| --- | --- | --- |
| China Antimicrobial Surveillance Network | 2020QD049 | Fupin Hu |
| Shanghai Antimicrobial Surveillance Network | 3030231003 | Fupin Hu |

## ADDITIONAL FILES

The following material is available online.

### Open Peer Review

**PEER REVIEW HISTORY (review-history.pdf).** An accounting of the reviewer comments and feedback.

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
