## [Reviewer comments · Microbiology Spectrum]

Microbiology Spectrum

Assessment of the revision of the 2025 CLSI breakpoints for the interpretation of minocycline susceptibility for *Acinetobacter baumannii* complex

Xiaochen Yu, Yanhua Liu, Jinye Du, Fupin Hu, and Dandan Yin

Corresponding Author(s): Dandan Yin, Institute of antibiotics

Review Timeline:

Submission Date:	September 11, 2025
Editorial Decision:	November 1, 2025
Revision Received:	December 11, 2025
Accepted:	December 21, 2025

Editor: Andrea Prinzi

Reviewer(s): Disclosure of reviewer identity is with reference to reviewer comments included in decision letter(s). The following individuals involved in review of your submission have agreed to reveal their identity: Ayesha Khan (Reviewer #1)

Transaction Report:

DOI: <https://doi.org/10.1128/spectrum.02700-25>

Re: Spectrum02700-25 (Assessment of the revision of the 2025 CLSI breakpoints for the interpretation of minocycline susceptibility for *Acinetobacter baumannii* complex)

Dear Dr. Dandan Yin:

Thank you for the privilege of reviewing your work. Below you will find my comments, instructions from the Spectrum editorial office, and the reviewer comments.

Revision Guidelines

Sincerely,
Andrea Prinzi
Editor
Microbiology Spectrum

Reviewer #1 (Public repository details (Required)):

Please include a supplemental file that includes the raw data so readers could examine the data themselves if needed. For example, an excel sheet with a list of strains (generic identifiers), BMD MICs per strain, zone diameters, interpretations of each, agree or disagree, column indicator errors or something similar. These files are very helpful for labs who need more granular data if they decide to validate a new method or when they update breakpoints.

Reviewer #1 (Comments for the Author):

I was pleased to review the manuscript entitled "Assessment of the revision of the 2025 CLSI breakpoints for the interpretation of minocycline susceptibility for *Acinetobacter baumannii* complex". The authors aimed to assess performance of the revised minocycline MIC and disk diffusion breakpoints set by CLSI in 2025 for *A. baumannii*. The study shows that the revised breakpoints improve error rates and correlation between disk and MIC results. While it is a single center study capturing isolates from a single country, the conclusions are informative and encourage labs to revise and update breakpoints to the new ones. See some comments that I hope may help improve the overall quality of the manuscript below:

1. Line 55: Specify the exact treatment regimen recommended by the IDSA guidelines. It may be helpful to include some background context or references on the evidence supporting the recommendation (or why it is preferred over other combination therapies for CRAB).
2. Line 58: elaborate briefly in 1-3 sentences what major performance issues have been reported for minocycline. The reference (#7) is not an open access article and the abstract alone does not relay major conclusions that would be pertinent to this study. Were performance issues summarized specifically for *Acinetobacter* testing or other organisms?
3. Line 61: Please include primary research article(s) as references here rather than a literature review
4. Clinical isolate selection: what criteria were used to select isolates? Are these invasive isolates? From any site or sterile sites only?
5. Line 87: What AST platforms were used to determine if isolates were CRAB or not? Were these results re-confirmed by the testing laboratory with a reference method?
6. Minor comment: Spelling of "here" needs to be corrected and is often spelled as "heer"
7. Line 163: Again, do these reviews recommend minocycline by itself or as part of a combination regimen? Clarify and be specific.
8. How do the MIC_{50/90} values reported in this specific study compare to those in published literature? Seems like the majority of CRAB isolates would be categorized as nonsusceptible, is this consistent with existing literature?
9. Line 165: What does "good in vitro activity" mean? Specify.
10. Line 168: What does hyper-virulence refer to? What criteria in general are used to categorize an isolate as hyper-virulent? This may not be common knowledge to readers.
11. Line 176: Minocycline-based combination therapy and which one? The preceding line mentions mino-polyB, it may be confusing if the exact treatment regimen you're referring to is not specified. What dosing?
12. Line 234: What do "clear results" mean? Easy to interpret? No trailing endpoints? Specify. Reproducible?
13. What does "irregular zone" specifically mean? I see Figure 4 but based on the images included, it is still unclear what makes these zones irregular. It seems like the overall lawn of bacteria can have slightly different appearances depending on the isolate but otherwise, visually at least, it looks like the zone diameter should be reasonably easy to measure? Maybe include more images or zoomed in panels with some qualitative descriptions in the results of what you mean when you say "irregular zones". Tetracyclines are bacteriostatic so some variation in growth is expected.
14. Line 243: Were the 3% of isolates showing "distortion" a specific MIC range? CRAB?
15. Minor comment: There's some grammatical and spelling mistakes throughout the manuscript. Make sure some form of auto-check proofreading is done in the resubmission to account for these. Nothing that detracts from the major conclusions so a minor point.
16. Line 262: In this section, it is still important to clarify that testing was performed at a single laboratory. The isolates were epidemiologically representative of *A. baumannii* circulating in China across regions but the study itself did not include multicenter testing.
17. Was the quality control strain tested with each run? With each lot of media? For both broth microdilution and disk diffusion? Clarify in section starting at line 88. Given previous reported variability with minocycline AST, important to specify quality control/assurance processes undertaken in the study.

Reviewer #2 (Comments for the Author):

The study evaluates the agreement between disk diffusion (DD) and broth microdilution (BMD) for the CLSI 2025 breakpoint update for *Acinetobacter baumannii* complex breakpoints for minocycline using 276 unique isolates with almost half of them being carbapenem resistant. The authors conclude the updated breakpoints significantly improve the agreement between these two methods (DD and BMD)

Major comments:

- 1) The CLSI denominator for the calculations for VME and ME should use the number of resistant isolates for VME and susceptible isolates for ME. This information can be found in the CLSI M52 Table A2 foot notes. Please make sure you make this clear in your data analysis section lines 101 - 104. Please double check that your calculations in the paper use this equation for your calculations.
- 2) The discussion needs to be written in a more specific and concise way. It is currently too long and not focused.
 - a. Line 162-163 is vague and does not add important information.
 - b. Line 165, please define what "good" means and is this data currently valid or does it use old breakpoints and is therefore not informative now?
 - c. Lines 165-168 (reference 17): Is study with old breakpoints or new breakpoints? Does the results interpretation of the paper change if the new breakpoints are applied?
 - d. Line 177: Please be specific in explaining where CRAB rates are up to 74%.
 - e. Line 180: Please explain what you mean by a "stepwise distribution".
 - f. Line 185: Please define "critical zone"
 - g. Please remove the line about reference 12 in lines 211-212. The information is not directly relevant to this study.
 - h. The last sentence in line 228 - 229 is unnecessary.
 - i. New figures should not be introduced in the discussion section. Please move the data and commentary about figure 4 to the main results section.
- 3) In Tables 2 and 3 please add the number of isolates in each category, not just the percentage.

Minor Comments:

- 1) Define CSAB in line 116
- 2) Figure 1: Please add headers for the 2025 and 2024 breakpoints at the top to make it easy for readers to follow.

Reviewer 1

1. Line 55: Specify the exact treatment regimen recommended by the IDSA guidelines.

It may be helpful to include some background context or references on the evidence supporting the recommendation (or why it is preferred over other combination therapies for CRAB).

Response: Thanks for your valuable suggestion. As outlined in the IDSA guidelines, the preferred treatment regimen for CRAB infections is sulbactam-durlobactam combined with either imipenem-cilastatin or meropenem. In cases where sulbactam-durlobactam is unavailable, an alternative option is high-dose ampicillin-sulbactam administered in combination with at least one additional active agent, such as polymyxin B, minocycline (prior to tigecycline), or cefiderocol. This information has been supplemented in the manuscript (Lines 68-73).

2. Line 58: elaborate briefly in 1-3 sentences what major performance issues have been reported for minocycline. The reference (#7) is not an open access article and the abstract alone does not relay major conclusions that would be pertinent to this study. Were performance issues summarized specifically for *Acinetobacter* testing or other organisms?

Response: Thanks for your comment. The reported performance issues of minocycline are specifically relevant to *Acinetobacter* testing and not applicable to other organisms. This section has been supplemented in the manuscript (Lines 76-81) with updated references (#7, 8) to provide sufficient supporting evidence.

3. Line 61: Please include primary research article(s) as references here rather than a literature review

Response: Thanks for your reminder. We have revised the references and replaced the literature review with primary research articles (#7, 8).

4. Clinical isolate selection: what criteria were used to select isolates? Are these invasive isolates? From any site or sterile sites only?

Response: Thanks for your question. Clinical isolates were selected based on our previous research data and bacterial strain bank. Strains were randomly chosen at a CR:CS ratio of 1:1, encompassing isolates from all anatomical sites (including non-sterile sites) and non-invasive strains.

5. Line 87: What AST platforms were used to determine if isolates were CRAB or not? Were these results re-confirmed by the testing laboratory with a reference method?

Response: Thanks for your inquiry. The broth microdilution method was used to determine the susceptibility of all isolates to imipenem (data not shown in the manuscript) for CRAB identification. Further clarification on this process has been added in Line 107.

6. Minor comment: Spelling of "here" needs to be corrected and is often spelled as "heer"

Response: Thanks for pointing out this error. We have corrected all instances of the misspelling.

7. Line 163: Again, do these reviews recommend minocycline by itself or as part of a combination regimen? Clarify and be specific.

Response: Thanks for your suggestion. This point has been clearly clarified in Lines 68-73 and Lines 201-203 of the manuscript. To avoid redundancy, the original sentence in the

Discussion section has been deleted.

8. How do the MIC50/90 values reported in this specific study compare to those in published literature? Seems like the majority of CRAB isolates would be categorized as nonsusceptible, is this consistent with existing literature?

Response: Thanks for your insightful question. As presented in Table 2, the MIC50/90 values for CRAB and CSAB in this study are 4/8 mg/L and 0.125/2 mg/L, respectively. According to Table 1 of the reference (Ann Clin Microbiol Antimicrob. 2025 Jan 13;24(1):2. doi: 10.1186/s12941-024-00766-4), the MIC50/90 values for 92 CRAB isolates in China were $\geq 8/16$ mg/L. When the MIC values from this reference are re-evaluated using the 2025 CLSI breakpoints, most isolates are still categorized as nonsusceptible, which is consistent with the findings of our study.

9. Line 165: What does "good in vitro activity" mean? Specify.

Response: Thanks for your comment. The phrase "good in vitro activity" originally intended to convey that CRAB isolates exhibited a relatively high susceptibility rate to minocycline in susceptibility testing, indicating that minocycline remains a viable alternative for combination therapy in CRAB treatment. Due to the lack of precision in the original expression, corresponding revisions have been made in Line 195.

10. Line 168: What does hyper-virulence refer to? What criteria in general are used to categorize an isolate as hyper-virulent? This may not be common knowledge to readers.

Response: Thanks for your question. Hyper-virulence refers to the enhanced pathogenicity of *Acinetobacter baumannii* strains, which is associated with more severe clinical outcomes. This clarification has been added in Line 197. As described in Reference #16, the *Galleria*

mellonella infection model was employed to assess the virulence level of strains, thereby classifying isolates into hypervirulent (hv-CRAB) and non-hypervirulent (non-hv-CRAB) groups.

11. Line 176: Minocycline-based combination therapy and which one? The preceding line mentions mino-polyB, it may be confusing if the exact treatment regimen you're referring to is not specified. What dosing?

Response: Thanks for your suggestion. Revisions have been made in Line 202. This section aims to illustrate that minocycline is an effective therapeutic option for *Acinetobacter baumannii* infections (either as monotherapy or in combination) based on relevant references. According to the reference, "Among the patients included, 223 (83.2%) received minocycline as monotherapy or combination therapy, while the remaining 45 (16.8%) were treated with other antimicrobial agents. For the 218 patients with available treatment data in the minocycline group, 18 (8.3%) received monotherapy and 200 (91.7%) received combination therapy. The standard dosage of minocycline in most studies was 100 mg twice daily (oral or intravenous formulation), with or without a loading dose of 200 mg. Exceptions were Studies E and F, where higher doses were administered (200 mg four times daily and 200 mg twice daily, respectively), although the authors did not provide sufficient justification for these higher doses. The most commonly used antimicrobials in combination with minocycline included colistin (intravenous or inhaled), polymyxin B, cefoperazone/sulbactam, and carbapenems (meropenem, doripenem, and imipenem ± cilastatin). "

12. Line 234: What do "clear results" mean? Easy to interpret? No trailing endpoints?

Specify. Reproducible?

Response: Thanks for your question. "Clear results" refer to unambiguous antimicrobial susceptibility testing results for minocycline. Specifically, in the broth microdilution method, there is no trailing growth, and in the Kirby-Bauer (KB) disk diffusion method, no beach-like growth or swarming growth is observed. These results feature clear endpoints that are easy to

interpret, enabling microbiology laboratory personnel to accurately and efficiently read and confirm susceptibility outcomes. Additionally, the results are reproducible.

13. What does "irregular zone" specifically mean? I see Figure 4 but based on the images included, it is still unclear what makes these zones irregular. It seems like the overall lawn of bacteria can have slightly different appearances depending on the isolate but otherwise, visually at least, it looks like the zone diameter should be reasonably easy to measure? Maybe include more images or zoomed in panels with some qualitative descriptions in the results of what you mean when you say "irregular zones". Tetracyclines are bacteriostatic so some variation in growth is expected.

Response: Thanks for your detailed comment. "Irregular zones" refer to areas of uneven bacterial growth observed in the Kirby-Bauer (KB) disk diffusion method, characterized by irregular shapes. For instance, as shown in Figure 4, the inhibition zone does not form a regular circle, resulting in variations in measured diameters at different positions. To facilitate understanding, the irregular regions have been marked in Figure 4. We acknowledge that tetracyclines are bacteriostatic, and some growth variation is expected. However, the irregular zones described herein exceed normal variation and may affect the accuracy of diameter measurement. Additional zoomed-in panels and qualitative descriptions of irregular zones have been supplemented in the Results section to enhance clarity.

14. Line 243: Were the 3% of isolates showing "distortion" a specific MIC range?

CRAB?

Response: Thanks for your question. The 3% of isolates exhibiting "distortion" were not confined to a specific MIC range nor exclusive to CRAB. This phenomenon may be related to random heterogeneity of the antibiotic disks. Among the 8 isolates with distortion, only one strain had an inhibition zone diameter falling within the critical range of 18-21 mm during measurement, and its susceptibility result requires confirmation via the broth microdilution method.

15. Minor comment: There's some grammatical and spelling mistakes throughout the manuscript. Make sure some form of auto-check proofreading is done in the resubmission to account for these. Nothing that detracts from the major conclusions so a minor point.

Response: Thanks for your reminder. We have carefully reviewed the entire manuscript and corrected all grammatical and spelling errors.

16. Line 262: In this section, it is still important to clarify that testing was performed at a single laboratory. The isolates were epidemiologically representative of *A. baumannii* circulating in China across regions but the study itself did not include multicenter testing.

Response: Thanks for your suggestion. Revisions have been made in Lines 232-236.

“Compared with previous single-center studies, the 276 *Acinetobacter baumannii* complex isolates included in our study were collected from multiple regions in China, exhibiting diverse geographic distributions and resistance backgrounds, which confers a certain degree of multicenter representativeness. However, it should be noted that all susceptibility testing was conducted at a single laboratory, and thus the findings of this study may have certain limitations.”

17. Was the quality control strain tested with each run? With each lot of media? For both broth microdilution and disk diffusion? Clarify in section starting at line 88. Given previous reported variability with minocycline AST, important to specify quality control/assurance processes undertaken in the study.

Response: Thanks for your important comment. Revisions have been made in Lines 112-118 to clarify this point. The quality control strain *Escherichia coli* ATCC 25922 was tested alongside clinical strains in each experiment, covering both the broth microdilution and disk diffusion methods (note: only one lot of media was used in this study). Additionally, a rigorous quality control and assurance protocol was implemented, including the adoption of standardized operating procedures (SOPs) and regular equipment calibration. Furthermore, the reading of experimental results in each assay was performed and cross-verified by two independent operators to ensure accuracy and reliability.

Reviewer #2

1. The CLSI denominator for the calculations for VME and ME should use the number of resistant isolates for VME and susceptible isolates for ME. This information can be found in the CLSI M52 Table A2 foot notes. Please make sure you make this clear in your data analysis section lines 101 - 104. Please double check that your calculations in the paper use this equation for your calculations.

Response: Thanks for your reminder. We have carefully reviewed the CLSI M52 guidelines and confirm that all VME and ME calculations in the manuscript adhere to the specified standards: the denominator for VME is the number of resistant isolates, and the denominator for ME is the number of susceptible isolates. This calculation method has been clearly specified in the data analysis section (Lines 126-128).

2. The discussion needs to be written in a more specific and concise way. It is currently too long and not focused.

Response: Thanks for your suggestion. We have thoroughly revised the Discussion section to enhance its specificity and conciseness, removing redundant content and focusing on key findings and their implications.

3. Line 162-163 is vague and does not add important information.

Response: Thanks for your comment. In response to Reviewer #1's feedback, the vague and non-essential sentence has been deleted.

4. Line 165, please define what "good" means and is this data currently valid or does it use old breakpoints and is therefore not informative now?

Response: Thanks. The phrase "good in vitro activity" originally intended to convey that CRAB isolates exhibited a relatively high susceptibility rate to minocycline in susceptibility testing, indicating that minocycline remains a viable alternative for combination therapy in CRAB treatment. The data presented was based on the old breakpoints and was indeed informative limited. Due to the lack of precision in the original expression, corresponding revisions have been made in Lines 194-195.

5. Lines 165-168 (reference 17): Is study with old breakpoints or new breakpoints?

Does the results interpretation of the paper change if the new breakpoints are applied?

Response: Thanks. Since the new CLSI breakpoints were only released in 2025, no published studies utilizing these new breakpoints for statistical analysis have been identified to date. If the new breakpoints were applied, the number of resistant isolates would increase. This further underscores the significance of the current study. According to our conclusions, the new breakpoints primarily impact CRAB isolates; therefore, re-evaluation of existing literature using the original data and new breakpoints is necessary.

6. Line 177: Please be specific in explaining where CRAB rates are up to 74%.

Response: Thanks for your suggestion. Revisions have been made in Line 204. According to Reference #20: "In a global analysis, the prevalence of CRAB isolates was found to be significantly higher (76.1% for imipenem, 73.5% for meropenem, 73% for doripenem, and 83.7% for ertapenem). Asia exhibited the highest incidence of carbapenem resistance at 76.2%. At the country level, resistance rates varied remarkably, ranging from 4% in Ireland to 96.1% in the Philippines."

7. Line 180: Please explain what you mean by a "stepwise distribution".

Response: Thanks for your question. We have revised the term to "gradual distribution" (Line 207) for greater accuracy. This refers to a continuous and progressive variation in the MIC values of tetracycline against different CRAB isolates, rather than clustering around a specific value.

8. Line 185: Please define "critical zone"

Response: Thanks for your comment. The "critical zone" refers to the MIC range adjacent to the susceptibility breakpoints. Due to methodological errors, the MIC values of the same strain may exhibit a ± 1 -dilution gradient shift across different tests. If the MIC value falls within this critical zone, it may lead to discrepancies in result interpretation.

9. Please remove the line about reference 12 in lines 211-212. The information is not directly relevant to this study.

Response: Thanks for your suggestion. We have deleted the line and Reference #12.

10. The last sentence in line 228 - 229 is unnecessary.

Response: Thanks for your comment. We've deleted.

11. New figures should not be introduced in the discussion section. Please move the data and commentary about figure 4 to the main results section.

Response: Thanks for your reminder. We have complied with the academic writing norms and moved the data and commentary related to Figure 4 from the Discussion section to the main Results section (Lines 184-190).

12. In Tables 2 and 3 please add the number of isolates in each category, not just the percentage.

Response: Thanks for your suggestion. We have revised Tables 2 and 3 to include the absolute number of isolates in each category alongside the corresponding percentages.

13. Define CSAB in line 116

Response: Thanks for your comment. CSAB (carbapenem-susceptible *Acinetobacter baumannii*) was defined as the isolates that susceptible to imipenem by broth microdilution, and this definition was shown in the preceding sentence (Line 101, 107).

14. Figure 1: Please add headers for the 2025 and 2024 breakpoints at the top to make it easy for readers to follow.

Response: Thanks for your suggestion. We have revised Figure 1 by adding clear headers for the 2025 and 2024 breakpoints at the top to improve readability for readers.

Re: Spectrum02700-25R1 (Assessment of the revision of the 2025 CLSI breakpoints for the interpretation of minocycline susceptibility for *Acinetobacter baumannii* complex)

Dear Dr. Dandan Yin:

Your manuscript has been accepted, and I am forwarding it to the ASM production staff for publication. Your paper will first be checked to make sure all elements meet the technical requirements. ASM staff will contact you if anything needs to be revised before copyediting and production can begin. Otherwise, you will be notified when your proofs are ready to be viewed.

Sincerely,
Andrea Prinzi
Editor
Microbiology Spectrum